# BAYESSHIFT: EVOLVING DOMAIN GENERALIZATION VIA HAMILTONIAN MONTE CARLO

## ABSTRACT

Evolving Domain Generalization (EDG) addresses learning scenarios where the data distribution evolves over time, a setting crucial for real-world applications under varying environmental conditions. Recently, structure-aware variational models have shown promise by disentangling static and variant information, but their reliance on point estimates for model parameters neglects parameter uncertainty, limiting both adaptability and reliability. We propose BayesShift, a full Bayesian framework that parameterizes a latent structure-aware autoencoder to capture static features, distribution drift, and categorical shifts. Unlike standard variational inference, our method leverages Hamiltonian Monte Carlo (HMC) to approximate the posterior over latent variables, enabling principled quantification of uncertainty, which not only improves robustness to evolving distributions but also provides confidence estimates for predictions, a critical property in safety-sensitive domains. Experiments on benchmark datasets demonstrate that BayesShift achieves higher robustness to evolving distributions, outperforming state-of-the-art baselines in both predictive accuracy and adaptability. These results highlight the effectiveness of Bayesian inference for evolving domain generalization.

## 1 INTRODUCTION

In many real-world machine learning applications, the assumption that training and test data share the same distribution often fails to hold, particularly in dynamic environments where conditions such as lighting, weather, or seasonal changes often evolve, leading to continuous shifts in data distribution. To address this challenge, **Evolving Domain Generalization (EDG)** has emerged as a promising direction. By explicitly modeling the progression of domain shifts, EDG aims to train models that noleverage the temporal or structural patterns of change (Nasery et al., 2021). This ability to adapt to evolving conditions is crucial for building more robust and generalizable systems in real-world applications.

Yet, existing EDG methods rely predominantly on *point estimates* of model parameters, which neglect uncertainty in representation learning. Existing approaches mostly focus on training deterministic neural networks that output single predictions without capturing posterior uncertainty over parameters or features (Zeng et al., 2023). As a result, they are often limited in adaptability and fail to provide reliable confidence estimates. To address these limitations, in this work we propose **BayesShift**, a full bayesian framework for evolving domain generalization. Our approach extends latent structure-aware autoencoders by integrating **Hamiltonian Monte Carlo (HMC)** to approximate the posterior over latent variables. Our method offers three key advantages:

- The structure-aware design disentangles static and dynamic components, while HMC enables more faithful tracking of the actual probability distribution modeled under both distribution drift and categorical shift.

- By capturing posterior variability, BayesShift provides confidence estimates for predictions based on uncertainty quantification, which is a critical property in safety-sensitive domains.

- Empirical evaluations across six benchmark datasets including both synthetic and real-world data show that BayesShift outperforms state-of-the-art methods.

## 2 BACKGROUND AND RELATED WORK

*Domain Generalization* (DG) aims to train models that generalize to unseen domains without access to target data, typically by learning domain-invariant features (Blanchard et al.; Li et al., 2018a; Albuquerque et al., 2019), using meta-learning (Li et al., 2017; Dou et al., 2019), or applying data augmentation (Volpi et al., 2018; Zhou et al., 2021). However, these methods often assume source domains are sampled from a static environment, limiting their effectiveness in dynamic settings. Continuous Domain Adaptation (CDA) addresses evolving distributions with intermediate domains (Kumar et al., 2020), domain manifolds (Hoffman et al., 2014), or adversarial learning (Wulfmeier et al., 2018), but typically requires access to target samples.

In contrast, *Evolving Domain Generalization* (EDG) tackles a more realistic scenario with no target data, leveraging environmental progression instead. Recent work on sequential data generation (Yingzhen & Mandt, 2018; Park et al., 2021) informs our approach by highlighting the value of disentangling time-varying and invariant features, though prior methods often ignore label-related information crucial for DG. However, current EDG methods rely predominantly on *p*oint estimates of model parameters. Approaches such as FORESEE uses transformers to simulate unseen target features (Zeng et al., 2023) and (Qin et al., 2022) uses VAE-based framework incorporating variational inference which gives an over-approximated result of latent structures. Existing methods focuses on implementing deterministic neural networks that output single point predictions, failing to reliably model the entire picture of data or domain features.

Moreover, *Markov Chain Monte Carlo*-based (MCMC-based) generative frameworks like Cycle-CoopNets (Xie et al., 2021) have employed alternating MCMC teaching to solve unsupervised cross-domain translation problems. Although not designed for evolving domains, this demonstrates the power of MCMC-based posterior sampling in cross-domain modeling. However, these works focus on aligning static domain shifts, without modeling progressive domain evolution over time. Thus, while MCMC have shown promise in domain-related tasks, their application to dynamic EDG remains largely unexplored. Our work aims to bridge this gap by adapting Bayesian sampling frameworks to track evolving environments.

## 3 METHODOLOGY

### 3.1 PROBLEM FORMULATION

The overall problem formulation is described as follows using the same terminology as (Wang et al., 2025). The input of the problem consists of $T$ continuous domains $\mathcal{S} = \{\mathcal{D}_1, \mathcal{D}_2, \ldots, \mathcal{D}_T\}$, each with labeled dataset $\{(x_t^n, y_t^n)\}_{n=1}^{N_t}$ drawn from evolving source distribution $P_t(X, Y)$. The goal is to learn a generalizable prediction function $f : \mathcal{X} \to \mathcal{Y}$ for target domains $\{\mathcal{D}_{T+1}, \mathcal{D}_{T+2}, \ldots\}$ along the direction of the underlying evolving domain shifts among the observed domains.

### 3.2 PROBABILISTIC GENERATIVE MODEL

To model the dynamics in the non-stationary systems, we consider the distribution drift in data space $W$ and shift in data category space $V$ respectively as proposed by Qin et al. (2023). For data $(x_t, y_t)$ at time $t$, we use latent variables $z_t^w$ and $z_t^v$ for each time stamp $t$, and time-invariant latent variable $z^c$ for static information in data over time, which are shown in Fig. 1. For the dynamic latent variables, $z_t^w$ accounts for the data distribution drift, and $z_t^v$ models the data category drift (also known as concept shift). Thus the joint distribution of all $T$ source domains can be written as:

$$p(x_{1:T}, y_{1:T}, z^c, z_{1:T}^w, z_{1:T}^v) = p(x_{1:T}, z_{1:T}^w, z^c)p(y_{1:T}, z_{1:T}^v|z^c). \tag{1}$$

We then use Markov Chain model to further derive the first and second term in 1:

$$p(x_{1:T}, z_{1:T}^w, z^c) = p(z^c) \prod_{t=1}^{T} p(z_t^w|z_{<t}^w)p(x_t|z^c, z_t^w), \tag{2}$$

$$p(y_{1:T}, z_{1:T}^v|z^c) = \prod_{t=1}^{T} p(z_t^v|z_{<t}^v)p(y_t|z^c, z_t^v), \tag{3}$$

where our goal is to learn model parameters $\theta$ of $p_\theta(z^c)$, $p_\theta(z_t^w|z_{<t}^w)$ and $p_\theta(z_t^v|z_{<t}^v)$.

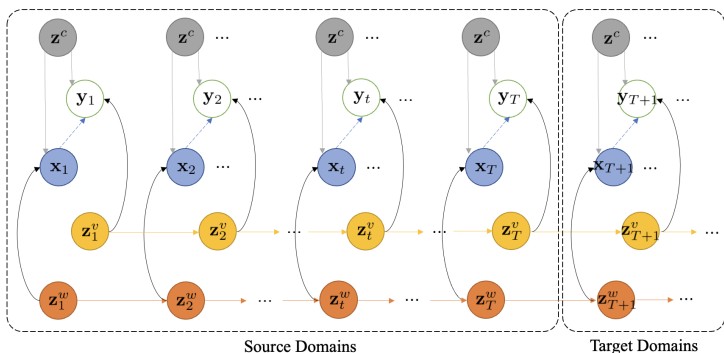

Figure 1: Diagram for our probabilistic generative model under a non-stationary domain generalization scenario where there exist evolving patterns among adjacent domains. The solid arrow lines stand for a dependency between variables (which is shown as conditional probability distributions in Equation 1). The dashed arrow line from input $x$ to label $y$ is not directly modeled in our framework.

### 3.3 HAMILTONIAN MONTE CARLO AS POSTERIOR ESTIMATOR FOR LATENT VARIABLES

Standard Variational Autoencoder (VAE) would use an Evidence Lower Bound Optimization (ELBO) for $\log p(x)$, instead we replace it with a fully Bayesian posterior estimator using Markov Chain Monte Carlo (MCMC) to sample latent posteriors. Particularly we choose Hamiltonian Monte Carlo (HMC) due to its higher acceptance rates and its use of gradient information in exploring the space. Other varients of MCMC methods including SGHMC (Chen et al., 2014) or SGLD (Welling & Teh, 2011), may vary in performance or convergence speed, and are also promising but not discussed in this paper.

The typical ELBO used in the variational inference process in VAEs is

$$\mathcal{L}_{\text{ELBO}} = \mathbb{E}_{q(z|x)}[\log p(x|z)] - D_{\text{KL}}(q(z|x)\|p(z)). \tag{4}$$

Instead of optimizing the ELBO (which introduces the variational $q$ and the KL), we approximate the marginal likelihood directly by sampling from the true posterior $p(z|x)$. A simple Monte-Carlo log-marginal estimate is:

$$\log p(x) \approx \log \int p(x|z)p(z)\, dz \approx \log \left( \frac{1}{N} \sum_{i=1}^{N} p(x|z^{(i)}) \right), \tag{5}$$

where $z^{(i)}$ is sampled via HMC. If the prior is standard Gaussian $p(z) = \mathcal{N}(0, I)$, $\log p(z) = -\frac{1}{2}\|z\|^2 + \text{const}$, by using Bayes rule we can derive:

$$\log p(z|x) \; \propto \; \log p(x|z) - \tfrac{1}{2}\|z\|^2. \tag{6}$$

Equivalently, writing negative log-likelihoods for $z^c$:

$$p(z^c|x) \propto \exp \Big( - \underbrace{\mathcal{L}_{\text{recon}}(x; z^c, z^w)}_{\text{negative log-likelihood}} - \underbrace{\tfrac{1}{2}\|z^c\|^2}_{\text{Gaussian prior}} \Big). \tag{7}$$

$z^w$ and $z^v$ follow the same form, note that we use a loss for classification on the negative log-likelihood of $z^v$ which is different from the reconstruction loss for the decoder of $z^c$ and $z^w$:

$$p(z^w|x) \propto \exp \Big( - \mathcal{L}_{\text{recon}}(x; z^c, z^w) - \tfrac{1}{2}\|z^w\|^2 \Big), \tag{8}$$

$$p(z^v|x) \propto \exp \Big( - \mathcal{L}_{\text{cls}}(y; z^c, z^v) - \tfrac{1}{2}\|z^v\|^2 \Big). \tag{9}$$

The HMC sampler seeks samples approximately from these posteriors by maximizing the log of reconstruction or classification term and prior. We calculate the gradient of the marginal log-likelihood

with respect to the decoder and classifier parameters $\theta$:

$$\nabla_\theta \log p_\theta(x) = \nabla_\theta \log \int p_\theta(x|z)\, p(z)\, dz = \frac{\int \nabla_\theta p_\theta(x|z)\, p(z)\, dz}{\int p_\theta(x|z)\, p(z)\, dz} = \mathbb{E}_{p_\theta(z|x)}\left[\nabla_\theta \log p_\theta(x|z)\right].$$

(10)

Thus we can approximate the gradient by Monte Carlo with $n$ samples $z^{(i)} \sim p_\theta(z|x)$:

$$\nabla_\theta \log p_\theta(x) \approx \frac{1}{n} \sum_{i=1}^{n} \nabla_\theta \log p_\theta(x|z^{(i)}),$$

(11)

which the gradients of the marginal log-likelihood equal posterior expectations of per-sample gradients. We approximate this by sampling $\{z^{c(i)}, z^{w(i)}, z^{v(i)}\}$ with HMC from their respective posteriors, then computing reconstructions $\hat{x}^{(i)} = p_\theta(x|z_c^{(i)}, z_w^{(i)})$ and classification logits $\hat{y}^{(i)} = p_\theta(y|z_c^{(i)}, z_v^{(i)})$, thus computing losses (negative log-likelihoods) and backpropagating them with respect to $\theta$. Therefore, the standard ELBO term is replaced by direct Monte Carlo (in particular HMC) estimates using posterior samples.

HMC employs Hamiltonian dynamics to traverse the parameter space, which the Hamiltonian is in form of $H(z, r) = U(z) + K(r)$. The kinetic energy term introduces a momentum variable $r \sim \mathcal{N}(0, M)$ (which is set to $\mathcal{N}(0, I)$ in our experiment), and the potential energy term is the negative log-posterior:

$$K(r) = \tfrac{1}{2} r^\top M^{-1} r, \quad U(z) = -\log p(z|x).$$

(12)

Gradients are computed via backpropagation through the decoder and priors. Samples are collected using HMC sampling using leapfrog integrator (Cobb et al., 2019), as shown in Algorithm 1.

---

**Algorithm 1** HMC sampling with leapfrog integrator for latent variable $z$

---

1: **Input:** Initial latent $z^{(0)}$, number of samples $n$, step size $\epsilon$, leapfrog steps $L$, log-posterior function $\log p(z|x)$, given data $x$
2: **for** $i = 1$ to $N$ **do**
3:      Sample momentum $r^{(i)} \sim \mathcal{N}(0, I)$
4:      Set $z^{(i)} \leftarrow z^{(i-1)}$, $r \leftarrow r^{(i)}$
5:      $r \leftarrow r + \frac{\epsilon}{2} \nabla_z \log p(z^{(i)}|x)$
6:      **for** $l = 1$ to $L$ **do**
7:         $z^{(i)} \leftarrow z^{(i)} + \epsilon \cdot r$
8:         Compute gradient $\nabla_z \log p(z^{(i)}|x)$
9:         **if** $l < L$ **then**
10:           $r \leftarrow r + \epsilon \cdot \nabla_z \log p(z^{(i)}|x)$
11:         **end if**
12:      **end for**
13: **end for**
14: $r \leftarrow r + \frac{\epsilon}{2} \nabla_z \log p(z^{(i)}|x)$
15: Compute accept-reject ratio $\alpha = \min\left(1, \exp\left[H(z^{(i-1)}, r^{(i)}) - H(z^{(i)}, r)\right]\right)$
16: Draw $u \sim \mathcal{U}[0, 1]$
17: **if** $u < \alpha$ **then**
18:      $z^{(i)} \leftarrow z^{(i-1)}$ /* Accept */
19: **end if**
20: **return** $\{z^{(1)}, \ldots, z^{(n)}\}$

---

### 3.4 Network Implementation

Our architecture follows the autoencoder structure proposed by Qin et al. (2023) as shown in Figure 2. $E^W$ and $E^C$ have the same architecture, which is implemented by a feature extractor. $D$ reconstructs $x$ from the latent variables. $E^V$ takes one-hot labels $y$ as input to produce categorical latent codes. The classifier $C$ is implemented by basic linear layers. We perform HMC sampling for latent variables $z$ in order to approximate their posterior probability function.

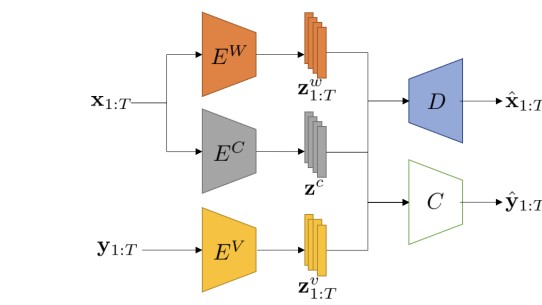

Figure 2: Network architecture consisting of encoder $E^W$, $E^C$, $E^V$, decoder $D$, and classifier $C$.

## 4 EXPERIMENTS

### 4.1 EXPERIMENTAL SETUP

We compare the performance of our framework with state-of-the-art domain generalization baselines, including ERM (Vergara et al., 2012), Mixup (Yan et al., 2020), MMD (Li et al., 2018b), DIVA (Ilse et al., 2020), and MMD-LSAE (Qin et al., 2023). To ensure fair comparison, we follow the same experimental protocol from Qin et al. (2022; 2023), where evolving domains are split into *source*, *intermediate*, and *target* domains. Since datasets used and testing procedure are identical to the previous work, we use partial experimental results whereas retesting the novel best-performing MMD-LSAE proposed. All models are trained on source domains, validated on intermediate domains, and tested on target domains that simulate unseen environments.

A summary of the datasets is provided below. Each dataset simulates evolving conditions, ranging from synthetic controlled shifts to complex real-world domain drifts.

1. **Circle** (Pesaranghader & Viktor, 2016): A synthetic benchmark with 30 evolving domains generated from 2D Gaussian distributions, forming in total 30 domains. Each subsequent domain introduces gradual drift in mean and covariance, testing the model's ability to capture continuous distributional changes.

2. **Sine** (Pesaranghader & Viktor, 2016): A synthetic benchmark of samples drawn from sinusoidal functions with evolving phase shifts and noise levels, with 24 domains in total. This dataset emphasizes the ability to model smooth but non-linear distributional evolution.

3. **RMNIST** (Ghifary et al., 2015): Contains real-world MNIST handwritten digit images rotated from $0°$ to $180°$ in $15°$ increments, forming 19 domains. The gradual increase in rotation angle simulates distribution drift common in visual tasks.

4. **PowerSupply** (Dau et al., 2018): A real-world time series dataset of electricity consumption. Domains evolve according to seasonal patterns and daily consumption cycles, introducing non-stationary dynamics, forming 30 domains according to the dates.

5. **Caltran** (Hoffman et al., 2014): A real-world traffic camera dataset capturing highway scenes under evolving weather and illumination conditions, forming 34 domains based on the change of time.

6. **Portraits** (Ginosar et al., 2015): A real-world facial portraits dataset from 1905s to 2005s, forming 34 domains based on the change of years.

Since the training design varies between different datasets, details of the training hyperparameters including model architecture, learning rate for each module, batch size, number of samples, number of leapfrog steps, and step size for numerical integration can be found in the code repository.

### 4.2 QUANTITATIVE RESULTS

Table 1 summarizes the average target domain accuracy across six benchmarks. Overall, BayesShift (ours) achieves the highest accuracy on four out of six datasets and the best overall average, outperforming existing domain generalization baselines by a substantial margin.

Table 1: Comparison of the average target domain accuracy (%) between our work and other DG baselines on various datasets.

| Algorithm | Circle | Sine | RMNIST | Portraits | Caltran | PowerSupply | Avg |
|---|---|---|---|---|---|---|---|
| ERM | 49.9 | 63.0 | 43.6 | 87.8 | 66.3 | 62.1 | 62.1 |
| Mixup | 48.4 | 62.9 | **44.9** | 87.8 | 66.0 | 62.0 | 62.0 |
| MMD | 50.7 | 55.8 | 44.8 | 87.3 | 57.1 | 59.1 | 59.1 |
| DIVA | 67.9 | 52.9 | 42.7 | 88.2 | 69.2 | 64.2 | 64.2 |
| MMD-LSAE | 49.6 | **87.3** | 32.6 | 74.5 | 41.9 | 69.8 | 59.3 |
| BayesShift (ours) | **79.5** | 74.8 | 43.2 | **89.4** | **70.6** | **71.0** | **71.6** |

Notably, on Circle dataset BayesShift achieves 79.5% accuracy, surpassing the best baseline (DIVA, 67.9%) by 11.6%, demonstrating its effectiveness under dynamic but structured domain drift. On Sine, our model achieves 74.8%, a strong performance gain compared to most baselines, though below MMD-LSAE, indicating our posterior sampling approach remains robust under periodic domain changes. On RMNIST where non-linear feature distortions pose significant challenges, BayesShift delivers competitive performance comparable to other approaches (within 1.7% accuracy lack to the best result). On the real-world benchmarks Portraits and Caltran, BayesShift achieves the highest accuracy, indicating better generalization to complex visual shifts. On PowerSupply, our method yields the best performance 71.0% on generalization to unseen seasonal patterns.

## 4.3 ABLATION STUDY

To quantify the impact of Bayesian posterior inference via sampling on domain generalization, we conduct an ablation study by replacing the HMC sampler in our framework with a standard variational inference (w/o MCMC) approach, following the original LSSAE implementation (Qin et al., 2022). Table 2 reports the target-domain performance comparison between the two variants.

Table 2: Improvement on target domain accuracy of our model over same architecture without using HMC.

| Dataset | w/o HMC (%) | with HMC (ours) (%) | Improvement (+ / − %) |
|---|---|---|---|
| Circle | 74.1 | 79.5 | +7.3 |
| Sine | 52.4 | 74.8 | +42.7 |
| RMNIST | 35.3 | 43.2 | +22.4 |
| Portraits | 89.0 | 89.4 | +0.4 |
| Caltran | 57.0 | 70.6 | +23.9 |
| PowerSupply | 70.8 | 72.9 | +0.3 |
| **Overall** | 63.1 | 71.6 | **+13.5** |

Our results show that HMC-based posterior inference provides substantial improvements across all benchmarks. On average, incorporating HMC leads to a 13.5% absolute gain in the average target domain accuracy over the variational inference baseline, confirming sampling-based Bayesian inference plays a critical role in improving the robustness and generalization of our model.

On Sine dataset, our method achieves a dramatic +42.7% improvement, underscoring HMC's strength in capturing structured, periodic changes of the posterior that may be difficult for VI to approximate. On real-world datasets Caltran and RMNIST, which contains non-linear feature distortions and more complex domain transitions, HMC provides +23.9% and +22.4% gains respectively, suggesting that more accurate posterior modeling would significantly improve latent feature alignment under challenging shifts. On simpler benchmarks such as Circle with a more gradual distribution drift, we observe a solid +7.3% improvement. While improvements on Portraits (+0.4%) and PowerSupply (+0.3%) are relatively modest, performance remains consistently higher with HMC. It might be that these scenarios have more stable or less complex domain shifts, thus variational inference is sufficient, but HMC still offers small but measurable gains.

The performance advantage arises from HMC's ability to generate high-fidelity samples from the true posterior distribution without restrictive parametric assumptions. Unlike variational inference, which underestimates uncertainty due to its reliance on simple approximations, HMC captures multimodal and non-Gaussian posteriors more accurately. This richer uncertainty representation provides more robust latent features, improving the generalization to unseen targets across diverse domains.

## 4.4 UNCERTAINTY ANALYSIS

A key advantage of our Bayesian framework is its ability to quantify predictive uncertainty through posterior sampling of latent variables. We implement posterior sampling from the distribution $p(y|x)$ by performing multiple forward passes through the model with stochastic sampling from latent spaces on Circle dataset. We calculate the predictive entropy to quantify total uncertainty in predictions. From Figure 3(a) we can't see any traits in the model accuracy along the evolving domain shift, but in (b) and (c) we observe a wider confidence interval width and higher uncertainty in domains farther from the training distribution, aligning with intuitive expectations.

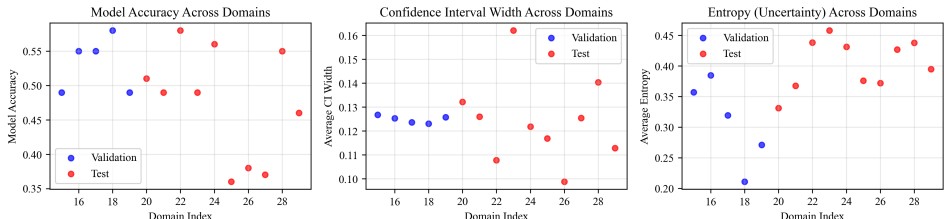

Figure 3: (a) Model accuracy, (b) width of confidence interval, and (c) uncertainty (modeled via entropy) across both intermediate domains (validation set) and target domains (test set) on Circle dataset.

The systematic increase in uncertainty from validation to test domains demonstrates that our model appropriately reflects its awareness of domain shift, providing reliable uncertainty estimates that can guide deployment decisions in real-world scenarios.

Moreover, we compute 95% confidence intervals using percentile-based estimation from posterior samples as shown in Figure 4, providing principled uncertainty bounds.

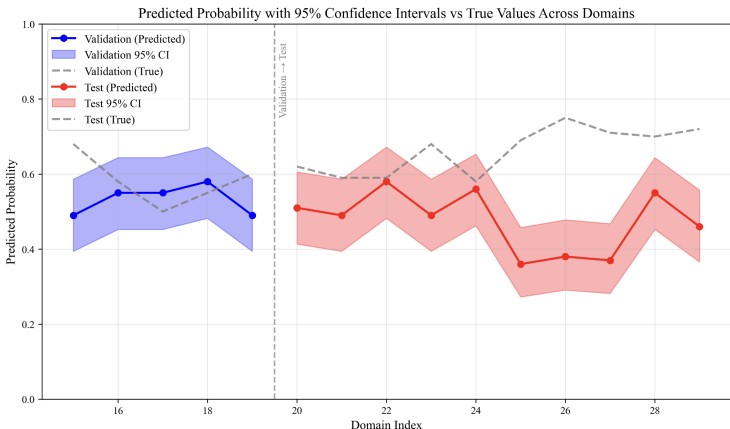

Figure 4: Confidence interval (95%) of the predicted value for Circle dataset, including the graph of each intermediate domain (validation set) and target domain (test set). The confidence bands demonstrate the model's uncertainty quantification capabilities, with wider intervals indicating higher uncertainty in test domains.

This uncertainty quantification capability is particularly valuable for safety-sensitive applications such as autonomous driving, where systems must not only predict accurately but also recognize

when they are uncertain. Our Bayesian framework provides principled uncertainty estimates that can inform decision-making processes and enable appropriate fallback mechanisms when model confidence is low.

## 5 CONCLUSION

In this work, we proposed a novel Bayesian framework for EDG, targeting the challenge of adapting a full Bayesian approach for machine learning models on gradually shifting data distributions. Our method extends the latent structure-aware autoencoder architecture by integrating HMC for posterior approximation, enabling more faithful modeling of both static and dynamic components in non-stationary environments. This integration allows the model to move beyond traditional point estimates or variational approximations, providing a principled Bayesian treatment of uncertainty that is essential for robust adaptation under evolving conditions. Empirical results demonstrate that our HMC-based approach improves generalization compared to conventional methods. Moreover, our framework lays a probabilistic foundation for uncertainty-aware domain generalization, offering interpretability and modularity that can facilitate downstream analysis.

However, several limitations remain. While HMC provides a more accurate approximation of the posterior, it is computationally expensive. Future work should explore alternative Bayesian sampling methods, such as SGHMC (Chen et al., 2014) and SGLD (Welling & Teh, 2011), which may offer better trade-offs between computational efficiency and posterior fidelity. Moreover, expanding the probabilistic generative backbone beyond the current autoencoder formulation could further improve representational capacity and adaptability in complex evolving domains. Applying MCMC sampling on other architechures such as diffusion model (Song et al., 2020) is a future promise. Also, evaluating the framework on a broader range of real-world, continuously shifting datasets is critical for assessing the model's practical utility. Integrating online or continual learning techniques with our Bayesian inference strategy also represents a promising direction for future research.

In summary, this work demonstrates that incorporating advanced Bayesian inference into EDG models significantly enhances their ability to capture posterior uncertainty and model evolving environments. By providing a principled framework for disentangling static and dynamic domain factors, our approach takes an important step toward robust, uncertainty-aware domain generalization in real-world, non-stationary settings.

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
