# OpenReview forum: "BayesShift: Evolving Domain Generalization via Hamiltonian Monte Carlo"
_ICLR.cc/2026/Conference — ICLR 2026 Conference Withdrawn Submission_

### Official Review · Reviewer_ny84 · 2025-10-29

**Soundness:** 4
**Presentation:** 3
**Contribution:** 2
**Rating:** 4
**Confidence:** 3

**Summary:**

This paper introduces BayesShift, a novel framework for Evolving Domain Generalization (EDG) that addresses performance degradation under evolving data distributions. The core idea is to replace the standard variational inference in a latent variable model with Hamiltonian Monte Carlo (HMC) sampling to achieve a more accurate posterior approximation. This approach aims to better disentangle static features from distribution and concept shifts, while also providing principled uncertainty quantification. The authors conduct experiments on several benchmarks, claiming superior accuracy over state-of-the-art methods and demonstrating the value of HMC through a strong ablation study.

**Strengths:**

1. The paper identifies a key limitation in current EDG research—the inadequate modeling of uncertainty. The introduction of HMC for posterior inference over latent variables in this context is a novel and theoretically sound contribution, providing a more principled Bayesian approach to handle the dynamics of evolving domains.

2. The paper demonstrates a key benefit of its Bayesian approach: principled uncertainty quantification. The experiments clearly show that the model's predictive uncertainty increases as domains shift away from the source distribution, highlighting its potential for building more trustworthy and reliable AI systems.

3. The paper is well-written and easy to follow. The authors clearly articulate the problem and position their work effectively. By building upon a known architecture, they successfully focus the reader's attention on their main innovation: the HMC-based inference mechanism.

**Weaknesses:**

1. The experimental validation (Table 1) for the model's claimed ability for distribution drift and categorical shift is insufficient. The paper lacks a granular analysis showing performance on two shift types, and the reported performance of a key baseline (MMD-LSAE) is suspiciously low compared to its original publication, which questions the fairness of the comparison and the proposed method's advantage.

2. The claim that uncertainty estimation is valuable for safety-critical applications is not substantiated with a concrete demonstration. The paper shows that the model can produce uncertainty estimates, but fails to provide a case study or experiment illustrating how these estimates could be used in practice to trigger a safety mechanism or improve decision-making.

3.  While the authors briefly mention the high computational cost of HMC in the conclusion, the evaluation requires a quantitative analysis of this critical trade-off. Please provide empirical data on key performance metrics such as inference latency, FLOPS, and number of parameters.

**Questions:**

Pls refer to weaknesses

---

### Official Review · Reviewer_fd71 · 2025-11-02

**Soundness:** 2
**Presentation:** 2
**Contribution:** 2
**Rating:** 4
**Confidence:** 4

**Summary:**

This paper introduces BayesShift, a Bayesian framework for Evolving Domain Generalization(EDG). It extends latent structure aware autoencoders by incorporating Hamiltonian Monte Carlo (HMC) sampling to estimate the posterior over latent variables, aiming to capture both static and dynamic domain features under evolving distributions. Theoretically, the method replaces the traditional variational inference (VI) used in prior DG work with a fully Bayesian sampling scheme. Experiments are conducted on six datasets (Circle, Sine, RMNIST, Caltran, PowerSupply, Portraits), and the authors claim that BayesShift achieves higher accuracy and robustness compared to several domain generalization baselines, including MMD-LSAE (an earlier EDG model). However, the experimental evaluation mainly compares against standard DG methods, and the only EDG baseline (MMD-LSAE) is not representative of current state-of-the-art EDG models such as SDE-EDG (ICLR 2024) or SYNC (ICML 2025). Therefore, while the HMC-based Bayesian formulation is technically sound, the evidence presented does not convincingly demonstrate progress in the EDG field.

**Strengths:**

•	Integrates Bayesian inference (via HMC) into domain generalization, providing uncertainty quantification.
•	Clear modular structure with disentangled latent variables for static and dynamic components.

**Weaknesses:**

•	Incomplete EDG benchmarking: only one EDG baseline (MMD-LSAE) is tested, which is not representative of current SOTA.
•	Lack of temporal modeling: the framework does not explicitly capture the dynamics between consecutive domains.
•	Limited novelty: substituting VI with HMC adds sampling fidelity but not conceptual progress for EDG.
•	Missing analysis: no discussion of HMC efficiency, convergence, or hyperparameter sensitivity.
•	Unclear theoretical justification: derivations are abbreviated and omit key intermediate steps.

**Questions:**

1. Why were newer EDG methods such as FORESEE, CTDG, or EvoS not included in comparison?
2. How sensitive is BayesShift to HMC hyperparameters (step size ε , leapfrog steps L, number of samples N)?
3. What is the computational cost relative to variational inference?
4. How does uncertainty estimation specifically help adaptation under evolving domains?

---

### Official Review · Reviewer_ckBD · 2025-11-03

**Soundness:** 2
**Presentation:** 3
**Contribution:** 2
**Rating:** 4
**Confidence:** 4

**Summary:**

The paper introduces BayesShift, a Bayesian framework for Evolving Domain Generalization (EDG) that augments a latent, structure-aware autoencoder with Hamiltonian Monte Carlo (HMC) to sample posteriors over latent variables capturing static content, distribution drift, and categorical shift. Replacing variational ELBO with HMC-based posterior sampling enables uncertainty quantification and, empirically, improves robustness to evolving domains. Across six benchmarks (synthetic and real), BayesShift attains the best average target-domain accuracy and provides calibrated confidence estimates; ablations indicate substantial gains from HMC over variational inference.

**Strengths:**

1. Models both distribution drift and concept shift with an explicit generative formulation, and uses HMC to capture posterior uncertainty rather than point estimates.
2. Provides predictive entropy and confidence intervals that increase with domain shift, aligning with safety-critical requirements.
3. Derives gradients as posterior expectations, details HMC with leapfrog integration, and integrates seamlessly with a structure-aware autoencoder.

**Weaknesses:**

1. HMC is expensive; the work lacks wall-clock/FLOP reporting, scalability analysis, and comparisons to efficient samplers (e.g., SGHMC/SGLD) or online/continual EDG settings.
2. Limited comparisons to recent EDG/CDA or Bayesian baselines; MMD-LSAE outperforms on Sine; tuning protocols and statistical significance (seeds, CIs) are under-reported.
3. HMC hyperparameters (step size, number of leapfrog steps), latent dimensionality, and encoder roles may strongly affect results; little analysis of sensitivity or disentanglement quality.

**Questions:**

Please refer to the weaknesses section.

---

### Note · Authors · 2025-12-01

**Comment:**

Thank you for all the reviews and constructive suggestions. We have taken all the comments seriously, but due to the limited time available for revision, we have decided to withdraw the paper for now. We will continue to refine and improve the work.

**Withdrawal Confirmation:**

I have read and agree with the venue's withdrawal policy on behalf of myself and my co-authors.